# 3D nanoprinting via spatially controlled assembly and polymerization

Thomas G. Pattison [1], Shuo Wang[2], Robert D. Miller[3], Gang-yu Liu [2✉] & Greg G. Qiao [1✉]

Macroscale additive manufacturing has seen significant advances recently, but these advances are not yet realized for the bottom-up formation of nanoscale polymeric features. We describe a platform technology for creating crosslinked polymer features using rapid surface-initiated crosslinking and versatile macrocrosslinkers, delivered by a microfluidic-coupled atomic force microscope known as FluidFM. A crosslinkable polymer containing norbornene moieties is delivered to a catalyzed substrate where polymerization occurs, resulting in extremely rapid chemical curing of the delivered material. Due to the living crosslinking reaction, construction of lines and patterns with multiple layers is possible, showing quantitative material addition from each deposition in a method analogous to fused filament fabrication, but at the nanoscale. Print parameters influenced printed line dimensions, with the smallest lines being 450 nm across with a vertical layer resolution of 2 nm. This nanoscale 3D printing platform of reactive polymer materials has applications for device fabrication, optical systems and biotechnology.

[1] Polymer Science Group, Department of Chemical Engineering, The University of Melbourne, Parkville, VIC 3010, Australia. [2] Department of Chemistry, University of California, Davis, CA 95616, USA. [3] International Business Machines—Almaden Research Center, 650 Harry Road, San Jose, CA 95120, USA. ✉email: gyliu@ucdavis.edu; gregghq@unimelb.edu.au

Three-dimensional (3D) printing has revolutionized manufacturing and prototyping[1–3], but its translation to the bottom–up, continuous formation of soft materials at the nanoscale remains a significant challenge primarily due to the difficulties in delivering the minute amounts of materials required with nanometer precision. A successful, robust 3D nanoprinting platform must address several critical design parameters such as a high degree of control over the spatial deposition of material, a continuous printing nature with minimal to no intermediate steps to stabilize printed layers without loss of feature registry between printing steps, and the deposition of solvent-free material to reduce or eliminate shrinking from solvent loss and enable the creation of multilayered features.

Herein, we report a 3D nanoprinting platform that enables the printing of polymer materials by design and with nanometer spatial precision. The approach combines the spatial precision of an atomic force microscope (AFM), the accurate materials delivery of a microfluidic probe[4–6], and the rapid curing using our solid-state continuous assembly and polymerization (CAP)[7]. A surface-initiated ring-opening metathesis reaction was used to bind a reactive ink, e.g., a solventless polymer, to the surface. This "ink" was delivered directly to the substrate using microfluidic AFM probes. With the surface functionalized by our chosen initiator and catalyst, the polymer was assembled and crosslinked immediately upon delivery. The exposed surface always remained functionalized due to the living nature of our chosen reactions. The amount of polymer delivered was controlled via our microfluidic AFM system by controlling the delivery conditions, e.g., pressure, speed, and contact time. The AFM allowed accurate movement by designed trajectory with nanometer precision. Therefore, this combination enabled the concept of 3D nanoprinting. In comparison with prior attempts of 3D nanoprinting using metallization[8–11] or molecular assembly[12–14], this approach can be applied to a wide range of polymer materials, exhibits high spatial accuracy and fidelity to design due to the rapid assembly and polymerization upon delivery. In contrast to conventional 3D printing which often uses thermal melting and solidifying during cooling[15–17], our reactions of curing occur at room temperature, with faster curing time and higher product stability.

To develop a nano-3D printing platform that could create polymer features in a continuous fashion, we used the Continuous Assembly of Polymers technique (CAP) to form stable structures. The CAP method uses polymer macrocrosslinkers containing crosslinking pendants that form highly crosslinked surface-tethered polymer films via surface-initiated polymerization (SIP), while maintaining a high degree of chemical tailorability via the remaining polymer side group functionality. This polymer is then delivered to a substrate initiated with surface-tethered catalyst molecules where the pendant monomer groups on the macrocrosslinker undergo SIP in the solid state[7], covalently binding the macrocrosslinkers to the surface. A variety of SIP methods[18,19] have been shown to work with CAP[20–22], but typically require similar degrees of deoxygenation and reagent purification to the analogous SIP in order to successfully form crosslinked material[23]. Ring-opening metathesis polymerization (ROMP) using the catalyst Grubbs Generation III has shown tolerance to moisture and oxygen[24], which is compatible with our 3D nanoprinting setup. However, even with the aforementioned advances, it is still difficult to perform commonly known surface-initiated ROMP successfully in air, as the catalyst typically undergoes degradation in ambient conditions[25,26]. To do so, our catalyzed surfaces were used immediately after initiation to prevent excessive degradation of the activated surface groups: substrates that have been modified with silane but yet to be exposed to Grubbs catalyst can be kept in inert atmospheres for at least up to 2 weeks and still yield crosslinked structures after activation with catalyst.

To achieve the microfluidic delivery of viscous material layer-by-layer[12], a new and highly reactive random co-polymer was designed (Fig. 1A, poly(polyethyleneglycol acrylate-co-hydroxyethylacrylate norbornene), p(PEGA-co-HEANB)) which undergoes ROMP-based crosslinking without the presence of solvent and in a rapid fashion before significant catalyst degradation. To do so, hydroxyethylacrylate and oligoethylene glycol monomethyl ether acrylate were copolymerized, and the product was functionalized post-polymerization to add highly reactive exo-norbornenyl pendants[27–29] to the co-polymer, resulting in the crosslinkable p(PEGA-co-HEANB). Acrylates and oligo/polyethyleneglycol-based materials have been widely used in a variety of biological applications, including the formation of crosslinked hydrogel materials[30–32]. The glass transition temperature of the polymer being $-52$ °C (Supplementary Figures 2–4) enabled the polymer to be printed as a viscous liquid at room temperature, avoiding issues with using solvent such as excessive material spreading and evaporation at the printing aperture. To facilitate the surface-initiated crosslinking upon delivery of polymer inks, quartz wafers were functionalized with [(5-bicyclo[2.2.1]hept-2-enyl)ethyl]methyldichlorosilane according to a previously published procedure[33], as illustrated in Fig. 1B. After functionalization, the wafers were exposed to a solution of 3rd generation Grubbs catalyst to anchor the metathesis catalyst to the surface, creating initiating sites for the ROMP reaction. The excess physisorbed catalyst was removed from the surface via repeated solvent washing before being blown dry with $N_2$. Reactivity of the synthesized ink was tested by spin-coating a solution of the polymer onto a silanized and initiated substrate (Fig. 1B) allowing the polymer to crosslink via surface-initiated ROMP. After 12 hours of reaction time, the film was washed to remove any un-crosslinked material, revealing a stable film 400 nm thick on the surface. Due to the living nature of the crosslinking ROMP reaction, further material can be added and is crosslinked by terminal catalyst species present at the surface of previously crosslinked polymer.

To spatially control the delivery of macrocrosslinkers (or polymer inks) to the pre-functionalized surface, an integrated AFM and microfluidic probe were used, referred to as FluidFM BOT (Fig. 1C). The neat p(PEGA-co-HEANB) liquid was loaded into the probe. The amount of material delivered to the surfaces was controlled by varying the probe-surface contact (AFM load), microfluidic pressure, and printing speed (or transient contact time). The system enabled delivery volumes as small as 0.5 aL[5]. The movement precision reached 5 nm over 1 mm range[12]. As a minute amount of p(PEGA-co-HEANB) was delivered to the designed location, the CAP crosslinking occurred immediately to solidify the ink, while the catalyst migrated to the outmost surface of the printed feature (Fig. 1B, C) to enable living CAP during subsequent printing. These observations demonstrate that the printing of multiple layers was accomplished without the need to reinitiate the previously deposited material—as no loss of catalyst activity was observed for at least 70 mins after deposition on surfaces in ambient conditions. Once lines were deposited and printing was finished, the substrates with printed structures were rinsed with dichloromethane (DCM) with 5% ethyl vinyl ether (EVE, to terminate the crosslinking), placed in 5% EVE in DCM for 1 hour, and then washed again with fresh DCM before placing under vacuum overnight prior to analysis via AFM.

## Results and discussion

**Surface-initiated 3D nanoprinting.** The concept of our 3D nanoprinting based on CAP was first demonstrated by printing an array of three polymer lines on a ROMP-active quartz surface. The lines were parallel with interline separation of 5 μm. As shown in Fig. 2A, these lines were identical in size and measured

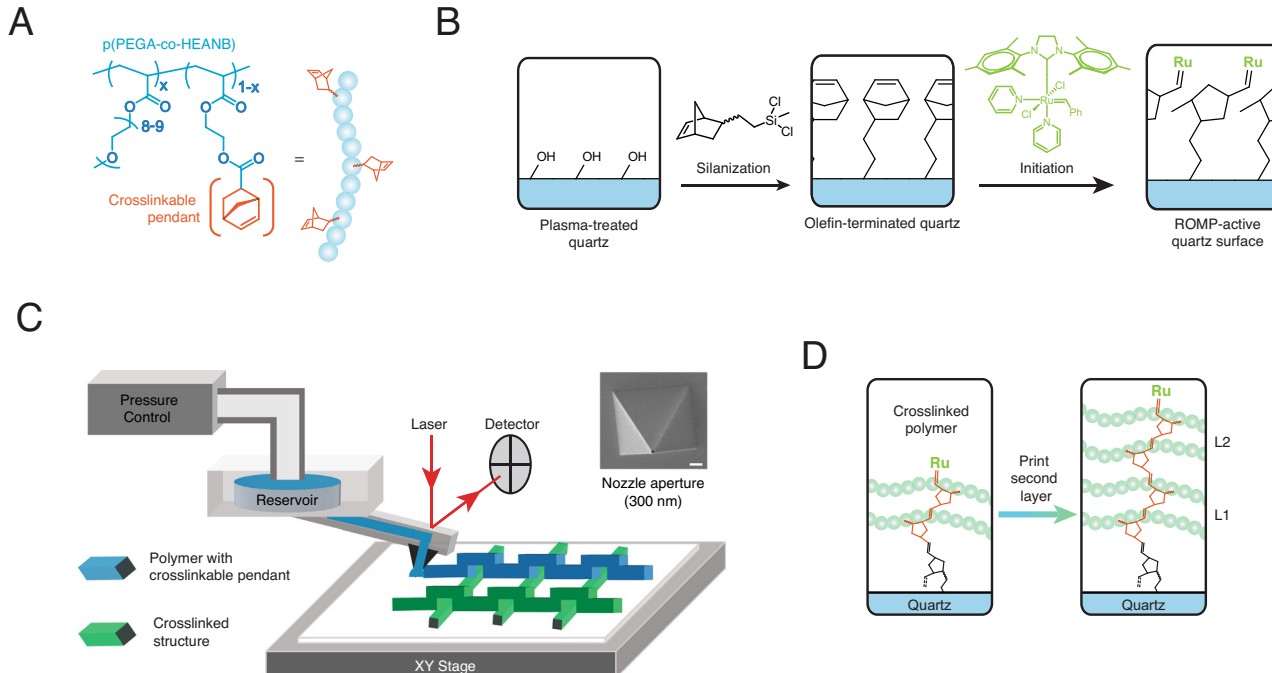

**Fig. 1 Schematic diagrams illustrating key steps in our 3D nanoprinting, including chemical crosslinking ink, surface modification and initiation, and the delivery of reactive ink to substrates to form 3D printed patterns. A** Structural formula of the crosslinking polymer, poly(polyethyleneglycol acrylate-co-hydroxyethylacrylate norbornene (p(PEGA-co-HEANB)), highlighting the crosslinkable pendant groups in orange. **B** Pre-functionalization of the surfaces with olefin-terminated SAM and initiated with the ROMP (ring-opening metathesis polymerization) catalyst Grubbs Generation III to enable Continuous Assembly of Polymers (CAP) upon delivery of p(PEGA-co-HEANB). **C** Schematic diagram illustrating the combined atomic force microscope (AFM) with microfluidic delivery, in realizing the CAP upon printing. The green material on the substrate shows crosslinked structures, while the blue is yet to be crosslinked. The inset scanning electron microscope image is of a FluidFM Nanopipette with a 300 nm aperture. Courtesy of Cytosurge AG. **D** Schematic of the resulting crosslinking formed after two layers of polymer have been deposited, showing the ability of the catalyst to continue crosslinking material. SEM scale bar: 2 μm.

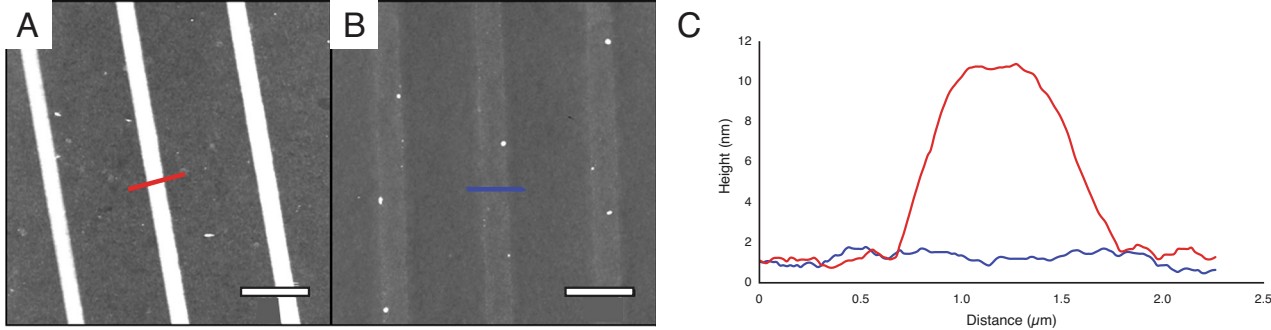

**Fig. 2 Atomic force microscopy images of 3D nano-printed line arrays using spatially controlled assembly and polymerization from initiated substrates as well as control substrates, demonstrating the active crosslinking reaction. A** 15 μm × 15 μm AFM topographic image of an array of the crosslinked polymer lines on a ROMP-active quartz surface. **B** 15 μm × 15 μm AFM topographic image of an array of lines printed with the same protocol as **A** without the catalyst present on a quartz surface. **C** Combined cursor profile from the two cursors indicated in **A** (red) and **B** (blue). Scale bars = 3 μm.

11 nm tall and 1.7 μm wide, and 100 μm long. Under identical conditions and using a clean quartz substrate without catalyst, control experiments were carried out. As shown in Fig. 2B, minimal p(PEGA-co-HEANB) assembly or cross-linking occurred. Figure 2C compares the AFM trace of the crosslinking system vs. the non-initiated control experiment (no catalyst), clearly showing the much taller line formed by the nano-3D printing system (11 nm) than that in the control (0.6 nm). The robustness of the observations was proven by printing larger features using a "micropipette" with 4 μm pore. An array of four lines, 300 μm long, were produced, each was 60 nm tall with the base width ranging from 10 to 25 μm (Supplementary Figure 5). In contrast,

control experiments did not yield any polymer lines. These tests confirm that assembly and catalyst-initiated crosslinking occurred during printing processes, i.e., spatially controlled CAP was the mechanism in our 3D nanoprinting.

**Influence of parameters on printed dimensions.** We could easily tune the printed line dimensions such as width and height by changing the printing parameters. Quantitative correlation between delivery parameters and liquid volume could be estimated using fluid dynamics approaches. A common example is derivations based on the Cox-Voinov equation using geometric

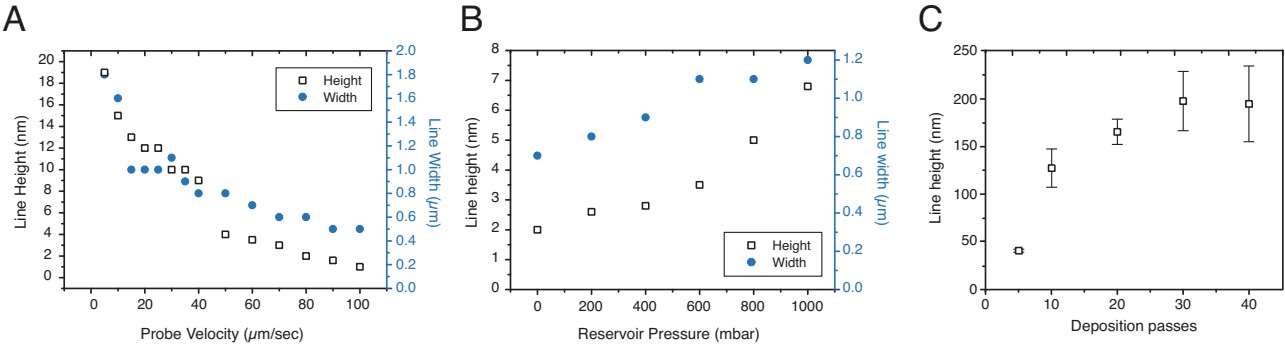

**Fig. 3 Data showing the influence of various printing parameters on the dimensions of the final printed lines. A** A plot showing the changes in the dimension of polymer lines with increasing printing speed, at 200 mbar. **B** A plot showing the variation of the dimension of polymer lines with increasing reservoir pressure, at the constant speed of 10 μm/sec. **C** Height of printed lines constructed after a designed number of passes and rinsing with a good solvent, error bars are st. dev. (where $n = 3$).

approximations[34,35]; another example is the energy balance to quantify the deposition process for macroscopic fluid delivery[36]. Before each experiment, we performed our own calibration[5] to correlate delivery parameters and delivered volume to address variations between individual tip apertures. The molecular assembly under spatial confinements was modeled and discussed in our prior work[5]. For materials used in this work, Fig. 3A, B shows that the line width and height could be varied by changing the printing speed and pressure. Under constant reservoir pressure, the extrusion rate of polymer is constant, thus the height of the line increased with the decreasing printing speed along the line, as demonstrated in Fig. 3A. Decreasing speed is equivalent to longer transient time, thus more material is extruded and available for crosslinking over a given distance. At 100 μm/sec and 200 mbar, the lines height and width were 1 nm and 400 nm, respectively. Much taller lines were produced by decreasing the printing speed—for example, at 5 μm/sec, lines are 19 nm tall and 1.85 μm wide. Increasing the printing pressure resulted in more material being extruded through the probe aperture per second and therefore a wider and taller line, as shown in Fig. 3B. Deposition was also observed at 0 mbar of reservoir pressure, likely due to the capillary action of the polymer through the aperture.

We also investigated the activity of the catalyst by depositing lines with an increasing number of passes, and then rinsing the surface to reveal stable polymer. When printing with the 300 nm aperture tip at 10 μm/sec and 1000 mbar of reservoir pressure, an increase in line-height was observed up to 30 depositions. Beyond this, the line height did not decrease. This corresponded with a line-height of 200 nm and indicates the burying of catalyst within the printed structure so that it can no longer crosslink further depositions without additional reinitiations. These results align with findings that were demonstrated with the CAP system in our previous work[33].

In comparison with optical-based 3D-printing methods such as two-photon direct laser writing, microstereolithography, and volumetric additive manufacturing (VAM), our approach, although slower in terms of printing speed, enables broader applications as we could print various materials without being restricted to using materials that contain photosensitive or photoactive molecules. Thus, this methodology could save time overall, as it reduces the time required to synthesize molecules containing photoactive functional groups, and in the preparation of photolithography such as formulating the crosslinking mixture and solubilizing photoinitiator, or pre-baking a photoresist layer. In addition, individual z-layer precision in this method is ~2 nm (single layer), which is much thinner (or high precision) than

what is typically achievable using optical-based 3D printing, these processes are usually limited by the diffraction limit (half of the wavelength). In comparison to other scanning probe microscopy (SPM) based nanolithography methods such as dip-pen nanolithography (DPN)[37−40], and nanografting[41,42], the spatial precision (in nanometers) is similar during the production of the first layer. The typical speed of SPM methods is 1−100 μm/s, and our approach can reach 1 mm/s[12]. These single probe methods have been scaled through the use of parallel arrays[43,44]. In terms of 3D nanoprinting by design, only our approach enables layer-by-layer delivery of materials by design.

**3D nanoprinting of structures**. The capability to print 3D structures by design was also demonstrated, with representative 3D structures shown in Fig. 4. The first structure is a set of stacked grids: an array of 10 parallel lines, 100 μm long with 10 μm separation, were printed atop a pre-functionalized quartz surface. Then the same printing protocols were followed by delivering 10 lines of p(PEGA-co-HEANB) atop and perpendicular to the first array. As shown in Fig. 4A the outcome exhibited high fidelity when compared to the design, with completely parallel lines separated by designed periodicity and perpendicularity from one array to the next without distortion or fuzzy edges. The height of lines measured to be 26 and 34 nm, for the first and second arrays, respectively, as shown in Fig. 4B. The cross-section was 60 nm tall, consistent with the concept of stacking individually printed passes on top of one another and is shown more clearly in the 3D image in Fig. 4C. The printing of multiple layers of a square pattern one after another is shown in Fig. 4D. The structure was created by printing 10 consecutive squares atop each other with 30 μm side length. The resulting structure shows a 3D square with walls of $98 \pm 12$ nm in height and $2.3 \pm 0.3$ μm in width, showing that the continuous deposition of materials enables the formation of taller features when compared to a single deposition, with minimal distortion in the final structure. Another 3D structure demonstrated was the stacking of cuboids (Fig. 4E) atop one another, whose designed dimensions are summarized in Table 1. The base was a cuboid with designed dimensions of 40 μm × 40 μm × 40 nm. Stacking atop this cuboid was a smaller cuboid with 20 μm × 20 μm × 15 nm. At the top were four cones designed to be 15 nm tall and 400 nm wide. As shown in Fig. 4E and outlined in Table 1, the outcome faithfully followed the design. The base cuboid measured as $42.1 \pm 1.6$ μm × $41.2 \pm 1.5$ μm in lateral dimensions and $61 \pm 5$ nm tall, produced by printing 133 lines at 300 nm separation. The next cuboid atop of the center region of the base measured $21.5 \pm 1.3$ μm × $20.8 \pm 1.4$ μm × $15 \pm 2$ nm created by

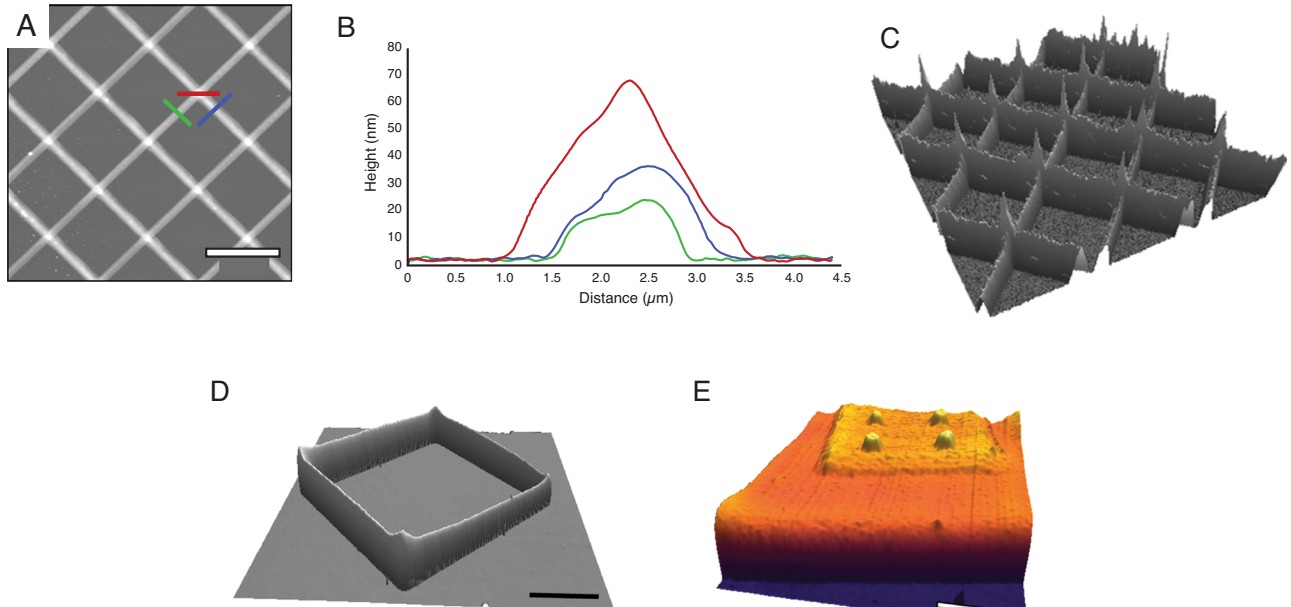

**Fig. 4 Atomic force microscopy of 3D nanoprinting following designed structures, highlighting the ability to form multilayered patterns atop of printed material. A** 40 µm × 40 µm AFM topographic image of the printed cross-grids of crosslinked polymer. **B** Combined cursor profiles from three colored cursors indicated in **A**. **C** A 3D display of **A** with z scale 0–80 nm. **D** A 3D display of a 50 µm × 50 µm AFM topographic image of the 3D square structure with z scale 0–150 nm. **E** A 3D display of a 35 µm × 35 µm AFM topographic image of stacked cuboids with z scale 0–120 nm. Scale bar in **A**, **D**, and **E**: 10 µm.

**Table 1 Design and geometries of the printed feature in Fig. 4E.**

| Geometry | Designed width (µm) | Number of lines | Designed separation(µm) | Measured width (µm) | Measured layer height (nm) |
|---|---|---|---|---|---|
| Bottom cuboid | 40 | 133 | 0.3 | 42.1 ± 1.6 × 41.2 ± 1.5 | 61 ± 5 |
| Middle cuboid | 20 | 40 | 0.5 | 21.5 ± 1.3 × 20.8 ± 1.4 | 15 ± 2 |
| Top 4 cones | 0.4 | 4 | 8 | 0.34 ± 23 | 17 ± 2 |

printing 40 lines with 500 nm separation. The four cones are located at the center of the four quadrants atop the second cuboid. Each cone was produced by dispensing a droplet of liquid polymer with a contact time of 1 s under 200 mbar delivery pressure. The dimensions of the cones measured 340 ± 23 nm at the base and 17 ± 2 nm tall. The capability of printing 3D structures by design has been demonstrated by other designs and using a variety of probes (Supplementary Figure 5). In all cases, the neat polymer ink crosslinked rapidly and maintained its stability for a minimum of 7 days. Additionally, depositing atop of this structure through the delivery of neat polymer does not alter the cured structures underneath. These observations further prove that our approach and materials enabled 3D nanoprinting by design.

## Conclusion

In conclusion, we have demonstrated a generic 3D printing method for creating stable three-dimensional micro- and nanostructures of polymeric materials by design. Using a scanning probe microscopy-based technology, the reactive ink materials are directly delivered to the localized site following the designed trajectory. Through a rapid SIP to form crosslinks, the delivered polymer ink is cured rapidly on surface contact, resulting in high fidelity of the 3D design. The living nature of our crosslinking reactions allows for continuous printing without requiring an additional application of catalyst or processing, resulting in truly continuous material delivery in multiple layers. High spatial selectivity and fidelity are achieved by varying printing parameters and the real-time control of the AFM-

nanofluidic platform. This approach is of generic importance in 3D nanoprinting of materials in general. Work is in progress to explore the feasibility and accuracy in 3D printing of other polymer materials such as block co-polymers, antibacterial materials, and nano-architectured materials including star polymers. Work is in progress for further miniaturization and to enable the printing of overhanging features, which would require the curing rate to be faster than the rate of material delivery. The combination of smart polymer engineering with three-dimensional, bottom–up micro/nanofabrication fills the technology void of 3D nanoprinting, with potential for use in a myriad of applications such as stimuli-responsive optical coatings, customized polymer features in microfluidics such as actuators, and tailored polymer surfaces to study cell-material interactions.

## Methods

**Materials**. DCM (99.8%) and tetrahydrofuran (THF, 99.9%) were received from J.T. Baker and were passed through a solvent purification tower to remove oxygen and moisture before use. Water was obtained from a Millipore MilliQ filtration system. Silicon wafers (1" diameter with native oxide) and quartz wafers (1" diameter) were purchased from Virginia Semiconductor, VA. [[5-bicyclo[2.2.1]hept-2-enyl]ethyl]methyldichlorosilane (tech-95, endo/exo isomers, 95%) was obtained through Gelest and used as received. Exo-5-norbornene carboxylic acid (5-NB-COOH, 97%) was obtained from Sigma Aldrich and used as received. Poly-ethyleneglycol monomethyl ether acrylate (PEGA, 99.5%) was purchased from Sigma Aldrich and passed through a column of basic alumina to remove the inhibitor immediately before use. 2-Hydroxyethylacrylate (HEA, 98%) was distilled at reduced pressure immediately before use. Azobisisobutyronitrile (AIBN, 98%) was obtained from Sigma Aldrich and was recrystallized twice from methanol. Unless otherwise stated, all other compounds were used as supplied.

**Wafer cleaning**. Polished silicon wafers or transparent quartz wafers were cleaned of particulates by sonication in acetone for 20 mins and then exposed to a 45 s oxygen reactive ion etch (RIE) cycle, which removes any organic residue and exposes the native oxide silanol groups. The wafers were then immediately modified using the procedure below.

**Silanization of quartz substrates**. In an $N_2$-filled glove bag, clean, plasma-activated quartz wafers were exposed to a 2.8% solution of [(5-bicyclo[2.2.1]hept-2-enyl)ethyl]methyldichlorosilane in anhydrous pentane for 6 hours. The silanized wafers were then sonicated for 5 minutes each in the following solvents: toluene, toluene, toluene/acetone (1:1), acetone for a total of 20 mins. They were then either used immediately for ssCAP$_{ROMP}$ reactions or were stored under $N_2$ until use.

**Attachment of catalyst to silanized quartz wafer**. In an $N_2$ filled glove bag, a solution of 6.7 mg/mL of 3rd generation Grubbs catalyst was prepared in anhydrous THF. The solution was filtered through a 0.45 μm PFTE filter and a wafer silanized with the olefin-dichlorosilane was placed into the solution for 20 mins. The wafer was then rinsed once in anhydrous THF and twice in anhydrous DCM to remove the physisorbed catalyst before being exposed to polymer without delay.

**Solid-state CAP$_{ROMP}$ from catalyzed, silanized quartz surface**. To test the polymer for surface-initiated crosslinking, the material was dissolved in dry THF to make a solution of 10 mg/mL before being spin-coated at 1200 RPM for 30 seconds, then held at 3000 RPM for 30 seconds before being placed in air or under $N_2$ for the requisite amount of time.

**Quenching of the surface-initiated ROMP reaction**. Substrates were removed from the vacuum drying oven or printing stage and were rinsed with copious amounts of THF/DCM before being rinsed in a container with neat EVE to remove the catalyst. The wafer was then rinsed once again with DCM before being vacuum dried for analysis.

**Printing experiments**. The printing process was performed using a FluidFM BOT (Cytosurge, Glattbrugg, Switzerland) containing an inverted optical microscope (IX-73, Olympus America, Center Valley, PA, USA). The xy-stage (Fig. 1C) was mounted onto an inverted optical microscope to monitor the position and delivery, with a lateral movement range of 240 mm × 74 mm, and a precision of 5 nm. The z-movement is independent of lateral movement with 4 nm precision over 50 mm. The substrate was placed on the xy-stage, while the probe was mounted to the vertical assembly controlling z-movement. The FluidFM Nanopipette or FluidFM Micropipette (CYPR/001511, Cytosurge, Glattbrugg, Switzerland) used for printing was integrated with a microfluidic delivery system. Typical probe-surface contact during delivery is shown in Fig. 1C, where the contact force was measured and controlled via similar means as conventional AFM with a deflection configuration. The square pyramidal tip-tilted 11° from the surface normal. The cantilever was similar to conventional AFM probes, 200 μm long, 36 μm wide, and 1.5 μm thick. The spring constant was 2 N/m. A typical nanopipette has a 300 nm diameter pore located at the probe apex, connected to a microchannel within. The microchannel was connected to a small reservoir where the polymer resided, and a mechanical pump and control system enabled the application of pressures from −800 mbar to +1000 mbar with 1 mbar precision. To print, the nanopipette was first filled with polymer by applying a high pressure (i.e., 1000 mbar) to allow the material to flow through the nanofluidic channel and fill the hollow section within the cantilever and tip. Hamilton 7000 series syringes, 1 μL (Hamilton, Reno, NV, USA), were used to deliver the printing material into the instrument's reservoir. Unless otherwise stated, typical printing parameters are: contact force = 40 nN, probe moving velocity = 10 μm/sec, and printing pressure = 200 mbar. Once the printing was completed, wafers were rinsed in solvent to remove any uncross-linked material, dried with a flow of nitrogen, and placed under vacuum to remove residual solvent before being characterized via AFM.

**Atomic force microscopy**. AFM images were acquired using a deflection type configuration (MFP-3D, Oxford Instrument, Santa Barbara, CA, USA). Probes (AC240-TS, Olympus America, Central Valley, PA, USA) of 1.7 N/m spring constant and 57 kHz resonant frequency were used to characterize the geometry and size of the printed structures. The driving frequency was set at the fundamental resonance of the cantilever, 57 kHz, and the damping was set at 40%. Image processing and display were performed using either the MFP-3D software developed on the Igor Pro 6.20 platform or Gwyddion (open-source software, Czech Metrology Institute, Brno, Czech Republic).

## Data availability

The source data used in this study (used to generate Figs. 2C, 3A–C, and 4B) is available in the figshare database under accession code (https://doi.org/10.6084/m9.figshare.17991482.v1). Source data are provided with this paper.

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

## Acknowledgements

The authors would like to thank Mr. A. Tek at IBM for technical assistance in thermal polymer analysis, and Ms. Yunbo Zheng at UC Davis for helpful discussions, as well as Cytosurge AG for providing the SEM used in Fig. 1. T.G. Pattison would like to acknowledge the Australian Postgraduate Award and the University of Melbourne/IBM Ph.D. Scholarship for funding this research. This work is also supported by National Science Foundation (USA, CHE-1808829, GYL) and Australian Research Council's Discovery Program (DP17010432 and DP130101846, GGQ) as well as Future Fellowship Program (FT11010100411, GGQ).

## Author contributions

T.G.P., S.W., R.D.M., G.L., and G.G.Q. helped in conceptualizing the study. T.G.P. carried out organic synthesis and characterization. T.G.P. and S.W. carried out 3D nanoprinting experiments and AFM analysis. T.G.P., S.W., R.D.M., G.L., and G.G.Q. assisted in manuscript preparation and editing.

## Competing interests

The authors declare no competing interests.
