## [Peer Review File · Nature Communications]

REVIEWER COMMENTS

Reviewer #1 (Remarks to the Author):

Review of 3D Nanoprinting via Spatially Controlled Assembly and Polymerization

In this manuscript, the authors use FluidFM atomic force microscopy to precisely deliver small quantities of resin, which reacts by living polymerization to create a 3-dimensional solid. The reported method is interesting, however the present study does not fully examine the printing mechanisms nor put the method in the context of other high-resolution 3D printing techniques and spm fabrication methods.

The present study is very phenomenological. The authors should consider some more rigorous modeling of the printing process. What sorts of volume/time can be printed, etc?

The method seems to inherently produce very large aspect ratio structures (width:height = 1000:10 nm). The authors should discuss applications that could benefit from such aspect ratios, especially given that most printing methods seek to align lateral and z resolution.

The authors should compare speed and resolution to high resolution printing methods such as 2 photon DLW and micro-stereolithography.

The new method shares similarities with dip pen nanolithography, which has been used to print 3d polymer brushes. This should also be discussed.

Many of the key applications for high resolution polymer 3d printing are in biology. The authors should discuss the biocompatibility of their resin.

An important aspect of 3d printing methods is the ability to print overhanging features. This does not seem possible in the new method. The authors should discuss this limitation.

Line width axis in Fig 3a should be um not nm.

Reviewer #2 (Remarks to the Author):

This manuscript reports the combination of surface-initiated polymerization with an AFM-based microfluidic system to enable the deposition of 3D features with macroscale areas and nanoscale heights. The technique is similar to dip pen nanolithography, as fashioned by Mirkin and other groups several years ago. In fact, Mirkin's group did use dip pen nanolithography to deposit ROMP monomers onto catalytically patterned substrates to produce polymers as early as 2003, and others attached polymers to nanopatterned substrates by "grafting to" approaches. What appears to be new to me in the current approach is 1) a polymer with polymerizable side groups is used instead of a monomer and 2) the technique demonstrates the ability to build up material in multiple steps, akin to 3D printing. The polymer ink here is well designed to contain the cross-linkable moieties and also

have liquid-like characteristics for effective printing. The authors are correct in that there could be several applications to this approach, especially if a diverse array of appropriate polymer inks can be developed. There are some issues I note below that the authors should address in a revision:

- 1) Is the structure rinsed with a good solvent after deposition? The SI suggests that dichloromethane is used for rinsing in at least one experiment. Does the solvent remove any of the polymer, or is the surface polymerization able to accommodate all the polymer that is deposited?
- 2) For the multi-step depositions, is new catalyst deposited after each step? If not, the authors should comment on the relative activity and stability of the catalyst to foster multiple step depositions, as this would be an important conclusion. If new catalyst is deposited, that should be clearly indicated, as the text is not clear on this issue.
- 3) P. 3, Lines 50-51: What is the subject of the sentence? Missing "polymer"
- 4) P. 4, Line 79: I don't think the polymer is ever defined. Expand the acronym.
- 5) P. 6, Line 115: "surfaces" should be singular.
- 6) P. 6, Line 119: subject-verb disagreement

Reviewer #3 (Remarks to the Author):

This is an interesting manuscript that will appeal to the broad readership of Nature Communications. By combining microfluidics with functional polymeric building blocks and AFM-based printing the authors have developed a powerful strategy for 3D fabrication of nanoscale polymeric objects. I strongly support publication after the authors have addressed the following points.

- 1) A significant opportunity for this work that would further increase impact would be to illustrate 3D printing with a range of starting polymeric building blocks. I encourage the authors to consider two key extensions which would demonstrate the utility of polymeric building blocks for 3D printing:
1) by varying MW, the viscosity of the ink could be tuned to further control printing. Given the Qiao groups expertise in various controlled radical processes, these systems should be easily obtained. I also note that information about the Mn and dispersity of the NB functionalized is absent from the SI or manuscript - this should be included. Full characterization illustrating the mole percent of NB groups along the backbone is also needed - what is the lower limit for NB groups for reproducible and efficient printing with these systems.
- 2) Demonstrating printing with a functionalized second polymer would be a major advance. For example, attachment of a chromophore to the acrylate backbone would be trivial and the printing of a second layer visualized by a variety of different techniques. This ability to 3D print objects based on multiple polymer starting materials would enable a variety of different applications and answer a number of important points raised by this manuscript - for example - what is the activity of the Ru-

chain ends after printing of the first and subsequent layers, is the same efficiency in crosslinking observed with multiple layers - how many layers can be printed? The introduction of additional functional monomers containing XPS "active" groups (i.e. S) may further enhance characterization and illustrate the versatility of using a polymeric comonomer building block with 2 or more "functional" repeat units.

2) The authors should discuss in greater detail the stability of various surfaces during this process - it is not immediately apparent from the text or the SI what are the limits - can the surfaces be activated and then held for much greater than 20 minutes before printing? Related to this question is how long is the printing cycle for a large area structure. This would be useful context for the reader and may also help with further designs and in understanding how many layers of polymer can be printed before no further reactivity is observed? Can a structure be overprinted with the same or different polymer after storage under ambient conditions for a day or more and crosslinking observed?

3) Two minor points - while I am not a great fan of the term "macrocrosslinkers", if the authors do use this term, please be consistent with respect to "macro cross-linker" and other variations. I would also advise that the claim of "first" in line 35 be removed, it is not needed to illustrate the novelty of this approach.

4) Given the feature sizes, microscopy techniques (SEM/TEM) other than AFM may allow significant insight into the 3D printing process, surface roughness, interfaces between layers, etc.. I note some similarity with photoresists feature sizes and the experience of IBM in this area is world-class.

Craig J. Hawker

Reviewer 1

Overall Remark: In this manuscript, the authors use FluidFM atomic force microscopy to precisely deliver small quantities of resin, which reacts by living polymerization to create a 3-dimensional solid. The reported method is interesting, however the present study does not fully examine the printing mechanisms nor put the method in the context of other high-resolution 3D printing techniques and spm fabrication methods.

We thank the reviewer for recognizing the significance of this work and for his/her time in providing valuable feedback.

1. The present study is very phenomenological. The authors should consider some more rigorous modeling of the printing process. What sorts of volume/time can be printed, etc?

We have done modelling in the past on this process and have cited that paper as Ref 5 in the revised manuscript. The volume, as reviewer correctly pointed out, depends on printing conditions, including contact time. Our printing volume ranges from 0.1 pL to 1 nL in this work. We also added discussions pertaining to this citation on Page 7.

2. The method seems to inherently produce very large aspect ratio structures (width:height = 1000:10 nm). The authors should discuss applications that could benefit from such aspect ratios, especially given that most printing methods seek to align lateral and z resolution.

Although we did have example structures with high aspect ratio, our method is not restricted to only high aspect ratio structures, as the technology enables the stacking and printing of materials in the z-direction as per researchers' needs. These examples were chosen to highlight a key advantage of this approach, i.e., the high spatial precision along the z-direction. In the revised manuscript, text was also added on Page 8 to highlight the very high z-layer resolution offered by this method, which is currently not achievable using other printing methods on this scale. The concluding remarks include applications that could benefit from the 3D structures made. We hope the explanation and revision clarify this aspect up to reviewer's satisfaction.

3. The authors should compare speed and resolution to high resolution printing methods such as 2 photon DLW and micro-stereolithography.

In comparison with optical based 3D printing methods such as two-photon direct laser writing, micro-stereolithography and volumetric additive manufacturing (VAM), our approach, although slower in terms of printing speed, enables broader applications because we could print various materials without being restricted to using materials that require photosensitive or photoactive molecules. Thus, this methodology could save time overall, as it reduces the time required to synthesize molecules containing photoactive functional groups, and in the preparation of photolithography such as

formulating the crosslinking mixture and solubilizing photoinitiator, or pre-baking a photoresist layer. We have included this aspect in the introduction and on Page 8, in the revision.

4. The new method shares similarities with dip pen nanolithography, which has been used to print 3d polymer brushes. This should also be discussed.

The physics and mechanism in our method **intrinsically differ** from that of dip pen nanolithography (DPN), as DPN relies on the capillary neck present at the contact at which solid materials from the probe diffuse in the local solution and onto the surfaces. There is little control over concentration and delivery. Ours is direct writing with absolute control over material concentration and delivery. Technology wise, DPN is largely a 2D technology, as going back to the printed features and delivering 2nd layer was not possible. Ours enables direct delivery at designated locations, which has much higher degree of control, and better suited for 3D nanoprinting. Due to these **scientific and technological differences intrinsically**, we hope it is acceptable that we focus on introducing our new methodology instead of making a comparison to DPN.

5. Many of the key applications for high resolution polymer 3d printing are in biology. The authors should discuss the biocompatibility of their resin.

We agree and appreciate the reviewer's point. Text and references (30-32) were added to discuss this aspect on Page 4.

6. An important aspect of 3d printing methods is the ability to print overhanging features. This does not seem possible in the new method. The authors should discuss this limitation.

The reviewer has identified a very interesting technical challenge, which indeed has been in our minds since the beginning of the project. It is, in principle, possible to printing overhanging structures, as long as the curing reactions are faster than our material delivery. Towards this goal, active approaches could include using polymers with faster reaction kinetics, or higher glass transition temperature; or further cooling the substrate. These approaches would require a significant variation to the polymer system and the device set-up. Therefore, we remain highly interested and plan to pursue this challenge in the future. Text was added to reflect this discuss under conclusion.

7. Line width axis in Fig 3a should be um not nm.

Thank you for the catch. It has been corrected.

Reviewer 2

Overall Remark: This manuscript reports the combination of surface-initiated polymerization with an AFM-based microfluidic system to enable the deposition of 3D features with macroscale areas and nanoscale heights. The technique is similar to dip pen nanolithography, as fashioned by Mirkin and other groups several years ago. In fact, Mirkin's group did use dip pen nanolithography to deposit ROMP monomers onto catalytically patterned substrates to produce polymers as early as 2003, and others attached polymers to nanopatterned substrates by "grafting to" approaches. What appears to be new to me in the current approach is 1) a polymer with polymerizable side groups is used instead of a monomer and 2) the technique demonstrates the ability to build up material in multiple steps, akin to 3D printing. The polymer ink here is well designed to contain the cross-linkable moieties and also have liquid-like characteristics for effective printing. The authors are correct in that there could be several applications to this approach, especially if a diverse array of appropriate polymer inks can be developed. There are some issues I note below that the authors should address in a revision:

We thank the reviewer for recognizing the significance of this work and for his/her compliments. We respectively point out that there are intrinsic differences in science and technology between our approach and the DPN reported by Dr. Mirkin (please see our discussion under reviewer 1, point 4).

1. Is the structure rinsed with a good solvent after deposition? The SI suggests that dichloromethane is used for rinsing in at least one experiment. Does the solvent remove any of the polymer, or is the surface polymerization able to accommodate all the polymer that is deposited?

The polymer is highly soluble in dichloromethane, and so washing the surface with DCM will certainly remove any un-crosslinked polymer. This was demonstrated in the control experiments whereby polymer material delivered to surfaces that do not have any catalyst on the surface show the removal of polymer material. To make this point clear, we have added text to the manuscript on Page 5 in order to ensure that readers understand this processing step. We appreciate the reviewer pointing this out, so thank you.

2. For the multi-step depositions, is new catalyst deposited after each step? If not, the authors should comment on the relative activity and stability of the catalyst to foster multiple step depositions, as this would be an important conclusion. If new catalyst is deposited, that should be clearly indicated, as the text is not clear on this issue.

This is a key point and one of the principal advantages of our approach. **No new catalyst** is necessary after each step, because the previously deposited catalyst was able to migrate atop to the outmost surface and continued to crosslink subsequent depositions. Text was added to assure the clarify of this point on Page 5. We have also conducted experiments to explore the extent to which a single activation of the surface can foster continuous deposition passes, with a new sub-figure added to Figure 3, namely Figure 3c. This figure shows the deposition of material for a differing number of passes, and then

measures the overall line height after solvent washing. It shows the successful crosslinking of material up to 30 passes with the 300 nm probe, indicating that depositions up to this line height can be accommodated by a single surface initiation. Text was added on Page 8 to explain this figure in the context of this question.

3. P. 3, Lines 50-51: What is the subject of the sentence? Missing “polymer”

This has been corrected.

4. P. 4, Line 79: I don't think the polymer is ever defined. Expand the acronym.

This has been changed.

5. P. 6, Line 115: “surfaces” should be singular.

This has been corrected.

6. P. 6, Line 119: subject-verb disagreement.

This has been corrected.

Reviewer 3

This is an interesting manuscript that will appeal to the broad readership of Nature Communications. By combining microfluidics with functional polymeric building blocks and AFM-based printing the authors have developed a powerful strategy for 3D fabrication of nanoscale polymeric objects. I strongly support publication after the authors have addressed the following points.

We thank the reviewer for recognizing the significance of this technology and for his/her high compliments.

1. A significant opportunity for this work that would further increase impact would be to illustrate 3D printing with a range of starting polymeric building blocks. I encourage the authors to consider two key extensions which would demonstrate the utility of polymeric building blocks for 3D printing: 1) by varying MW, the viscosity of the ink could be tuned to further control printing. Given the Qiao groups expertise in various controlled radical processes, these systems should be easily obtained. I also note that information about the Mn and dispersity of the NB functionalized is absent from the SI or manuscript - this should be included. Full characterization illustrating the mole percent of NB groups along the backbone is also needed - what is the lower limit for NB groups for reproducible and efficient printing with these systems.

We totally agree that this approach has broad applications. We have added data pertaining to the polymer characteristics taken from GPC in the supplementary information. We have applied our early approach to various types of polymers, see Refs 5, 12, and plan to test more materials. Reviewer is correct that the Qiao team has access to a broad range of materials, and we do plan to continue this research direction for this approach. Since this work represents a proof of concept of this approach focusing on a specific class of materials (acrylates) as an example, which indicate the capability of the crosslinking reaction in this application, we hope this work reach the readers first. We also added this aspect in our concluding remark.

2. Demonstrating printing with a functionalized second polymer would be a major advance. For example, attachment of a chromophore to the acrylate backbone would be trivial and the printing of a second layer visualized by a variety of different techniques. This ability to 3D print objects based on multiple polymer starting materials would enable a variety of different applications and answer a number of important points raised by this manuscript - for example - what is the activity of the Ru-chain ends after printing of the first and subsequent layers, is the same efficiency in crosslinking observed with multiple layers - how many layers can be printed? The introduction of additional functional monomers containing XPS "active" groups (i.e. S) may further enhance characterization and illustrate the versatility of using a polymeric comonomer building block with 2 or more "functional" repeat units.

We agree that this approach could be enhanced by combining with other technologies and advanced chemical reactions. We appreciated the ideas and

discussion reviewer raised, which indicates the interests our initial work could sparkle. Again, these further explorations are best suited for our future work and publications.

To answer the question regarding Ru activity, we conducted printing experiments using a single initiation of the surface, and then printed multiple overlapping lines (up to 40) and then measured the thickness of the crosslinked polymer after rinsing. The sub-figure has been added to Figure 3 (Fig 3c), which shows that with a single initiation and printing with the 300 nm probe at 10 $\mu\text{m}/\text{sec}$ and 1000 mbar reservoir pressure, the maximum line height achievable is approximately 200 nm. Discussion to accompany the figure was added to Page 8.

3. The authors should discuss in greater detail the stability of various surfaces during this process - it is not immediately apparent from the text or the SI what are the limits - can the surfaces be activated and then held for much greater than 20 minutes before printing? Related to this question is how long is the printing cycle for a large area structure. This would be useful context for the reader and may also help with further designs and in understanding how many layers of polymer can be printed before no further reactivity is observed? Can a structure be overprinted with the same or different polymer after storage under ambient conditions for a day or more and crosslinking observed?

We functionalised surfaces immediately prior to printing – the Grubbs catalyst becomes susceptible to oxidation once initiation occurs and so the surface should be catalysed immediately prior to printing.

- For silanized surfaces, they can be kept in inert conditions for up to at least 2 weeks and still be active for initiation.
- For catalysed surfaces, they should be used immediately.
- For printed polymer structures, once activated the catalyst will degrade in ambient conditions after some time – they cannot be reprinted over after 1 day without further initiation.
- Re-initiation is possible by simply adding catalyst to react with unconsumed double bonds at the surface of formed structure.

Considering these questions, we have added a section on Page 4 specifying the information regarding stability of catalysed surfaces and their use for stable printing, as well as a short edit in the Methods section about using catalysed surfaces “without delay”. The new data added as Fig 3c also assists readers in determining the maximum possible feature printed before reinitiation is required.

4. Two minor points - while I am not a great fan of the term "macrocrosslinkers", if the authors do use this term, please be consistent with respect to "macro cross-linker" and other variations. I would also advise that the claim of "first" in line 35 be removed, it is not needed to illustrate the novelty of this approach.

This has been changed.

REVIEWERS' COMMENTS

Reviewer #1 (Remarks to the Author):

The authors have responded to my previous comments, but the responses are still largely phenomenological in nature. This was my primary criticism of the previous draft. The manuscript still fails to put the new method in a quantitative context of previous or competing methods. Namely, a quantitative comparison of 3D printing speed between this method and some existing methods is necessary. Stating that the method is slower is not sufficient. The reader should know if it is 2x slower or 10,000x slower. Throughput limitations have hindered adoption of previous SPM fabrication methods. Is the new method compatible with parallelized fabrication to address this limitation?

Likewise, the other reviewer and I saw the immediate connection to dip pen lithography. Some quantitative comparisons seem necessary, or at least comparison to other scanning probe based fabrication methods.

I fail to see how ref 5 (<https://doi.org/10.1021/acs.jpcllett.8b02442>) in the revised manuscript addresses the modeling concerns I outlined in my prior response.

I'd like to see the authors demonstrate the ~1:1 aspect ratio structures that they purport in their response.

Reviewer #2 (Remarks to the Author):

The authors have addressed my critical points well and included new text to provide greater clarity. I believe the manuscript is now stronger and suitable for publication.

Reviewer #3 (Remarks to the Author):

The authors have addressed all of the major points raised by the referees. The manuscript is now acceptable for publication.

Reviewer 1

The authors have responded to my previous comments, but the responses are still largely phenomenological in nature. This was my primary criticism of the previous draft. The manuscript still fails to put the new method in a quantitative context of previous or competing methods.

Namely, a quantitative comparison of 3D printing speed between this method and some existing methods is necessary. Stating that the method is slower is not sufficient. The reader should know if it is 2x slower or 10,000x slower. Throughput limitations have hindered adoption of previous SPM fabrication methods. Is the new method compatible with parallelized fabrication to address this limitation?

The peer technology of optics-based 3D nanoprinting has been compared with our approach, see page 8.

The principal advantage over photo-lithography, as stated in the manuscript, is the fact that our approach **removes the restriction or limitation** of using photo-sensitive molecules. This advantage broadens up the applications as more materials could be used in our method. The advantage of “being faster” is merely “an ice on the cake”, as stated, because the time required to incorporate photo-sensitive functional groups into molecules is no longer required. We do understand and appreciate that the reviewer emphasizes quantitative information, instead of merely stating one technology is slower or faster than the other. The reason we do not quantify this comparison is the fact that time varies tremendously from synthesizing one material to another in photon-based methods. If we take molecules in 3D photo-printing 1-by-1 for comparison of time, it would

[Redacted]

[Redacted]

definitively deviate from **the principal advantages and true reasons** to introduce this new approach.

Understanding the spirit of Nature, we hope the editor would agree with the principal advantages our technology offers, and the merit to introduce a new approach to readers instead of just staying with current technologies which have significant limitations in what materials could be used.

Likewise, the other reviewer and I saw the immediate connection to dip pen lithography. Some quantitative comparisons seem necessary, or at least comparison to other scanning probe based fabrication methods.

It is natural to see the immediate connection with SPM, as our instrument is built atop of SPM, “the shoulder of a giant,” so to speak. However, the end technologies intrinsically differ, as also explained in our previous responses. An analogy would be “Windows versus DOS” in PC operation system. The physics and mechanism in our method **intrinsically differ** from that of dip pen nanolithography (DPN), as DPN relies on the capillary neck present at the contact at which solid materials from the probe diffuse in the local solution and onto the surfaces. There is little control over concentration and delivery. Ours is direct writing with absolute control over material concentration and delivery. Technology-wise, DPN or scanning probe microscopy-based technologies are largely a **2D** technology, **not 3D printing by design**, as going back to the printed features and delivering 2nd layer is extremely difficult, and almost impossible, in DPN.

If forced to compare and quantify speed, DPN’s speed for 3D printing would be zero, and ours could reach 1 mm/s, **so the ratio would be infinite**, **Ours:DPN = 1 : 0 = ∞**. It would not make sense for this manuscript to make such a comparison and claim, because it is an “apple versus orange” situation. We also do not wish to put DPN down like that, as DPN is a valuable technology in its own right when forming 2D patterns.

Therefore, we took the reviewer’s message as “provide a comparison with DPN for the benefit of readers”. In this spirit, we included the above messages plus some quantitative discussions in the revised manuscript. Please note that DPN could only fabricate layer 1, while ours could perform layer-by-layer following the design. So we could only compare layer 1 to avoid “apple versus orange” scenario, see page 9 for added comments. We hope these changes would put this issue to rest.

I fail to see how ref 5 (<https://doi.org/10.1021/acs.jpcclett.8b02442>) in the revised manuscript addresses the modeling concerns I outlined in my prior response.

We apologize for our previous mis-understanding of the reviewer’s critiques. Based on the context of our manuscript and reviewer’s critiques, we thought that the reviewer was asking for our new modeling and molecular assembly on surfaces. Reading again, he/she was seeking more basic information, i.e., the microfluidic delivery, which is much more established than the topic in ref. 5. We included the two well-known quantification methods, namely the Cox-Voinov

equation using geometric approximations and the energy balance model to quantify the deposition process for macroscopic fluid delivery. Before each experiment, we also performed our own calibration to correlate delivery parameters and quantify to address the tip variations. Please see page 7 for more detailed revision plus three new citations, refs 34-36.

I'd like to see the authors demonstrate the $\sim 1:1$ aspect ratio structures that they purport in their response.

We believe the reviewer wishes to see a feature with lower aspect ratio than our reported one, i.e., 1000:10. Below are some structures we produced using the same 3D printing technology but different chemistry, which achieves the aspect ratio $\sim 2:1$, close to reviewer's needs. These data do not pertain to the focus of our work, so we only include them here in the letter to demonstrate the capabilities of our technology.

Figure: 3D nanoprining of a “chimney”-shaped assembly, and a sharp “cone” atop of a “pan-cake” structure. The chimney has aspect ratio of $\sim 600 \text{ nm} : 300 \text{ nm} = 2:1$, while the cone itself has aspect ratio of $\sim 1000 \text{ nm} : 200 \text{ nm} = 5:1$.